# CharXgen-Activated Bamboo Charcoal Encapsulated in Sodium Alginate Microsphere as the Absorbent of Uremic Toxins to Retard Kidney Function Deterioration

**DOI:** 10.3390/ijms21041257

**Published:** 2020-02-13

**Authors:** Cheng-Jui Lin, Chiao-Yin Sun, Chih-Jen Wu, Chau-Chung Wu, Vincent Wu, Feng-Huei Lin

**Affiliations:** 1Division of Nephrology, Department of Internal Medicine, Mackay Memorial Hospital, Taipei 10449, Taiwan; lincj@mmh.org.tw (C.-J.L.); yaliwcj@gmail.com (C.-J.W.); 2Mackay Junior College of Medicine, Nursing and Management, Taipei 11260, Taiwan; 3Department of Medicine, Mackay Medical College, New Taipei City 252, Taiwan; 4Division of Nephrology, Department of Internal Medicine, Keelung Chang Gung Memorial Hospital, Keelung 20401, Taiwan; fish3970@gmail.com; 5Division of Cardiology, Department of Internal Medicine, National Taiwan University Hospital, Taipei 10048, Taiwan; chauchungwu@ntu.edu.tw; 6Division of Nephrology, Department of Internal Medicine, National Taiwan University Hospital, Taipei 10048, Taiwan; 7Institute of Biomedical Engineering, National Taiwan University, Taipei 10048, Taiwan

**Keywords:** Bamboo charcoal, CharXgen, Chronic kidney disease, Indoxyl sulphate, P-cresolsulphate, Fibroblast growth factor 23

## Abstract

Indoxyl sulphate (IS) and p-cresyl sulphate (PCS) are two protein bound uraemic toxins accumulated in chronic kidney disease (CKD) and associated with adverse outcomes. The purpose of this study isto evaluate the effect of the new activated charcoal, CharXgen, on renal function protection and lowering serum uraemic toxins in CKD animal model. The physical character of CharXgen was analyzed before and after activation procedure by Scanning Electron Microscope (SEM) and X-ray diffractometer (XRD). The effect of CharXgen on biochemistry and lowering uremic toxins was evaluated by in vitro binding assay and CKD animal model. CharXgen have high interior surface area analyzed by SEM and XRD and have been produced from local bamboo after an activation process. CharXgen was able to effectively absorb IS, p-cresol and phosphate in an in vitro gastrointestinal tract simulation study. The animal study showed that CharXgen did not cause intestine blackening. Serum albuminand liver function did not change after feeding with CharXgen. Moreover, renal function was improved in CKD rats fed with CharXgen as compared to the CKD group, and there were no significant differences in the CKD and the CKD + AST-120 groups. Serum IS and PCS were higher in the CKD group and lower in rats treated with CharXgen and AST-120. In rats treated with CharXgen, Fibroblast growth factor 23 was significantly decreased as compared to the CKD group. This change cannot be found in rats fed with AST-120.It indicates that CharXgen is a new safe and non-toxic activated charcoal having potential in attenuating renal function deterioration and lowering protein-bound uraemic toxins. Whether the introduction of this new charcoal could further have renal protection in CKD patients will need to be investigated further.

## 1. Introduction

The risk of cardiovascular disease (CVD) is markedly increased in patients with chronic kidney disease (CKD) [1], especially in those with end stage renal disease (ESRD) [2,3]. Traditional CVD risk factors such as hypertension, hyperlipidaemia, and diabetes cannot fully explain the increased CVD risk in these patients [4,5]. Evidence has supported and emphasized the concept of non-traditional risk factors [6,7,8,9], which include accumulation of uraemic toxins, with a decline in renal function. Uraemic solutes, including indoxyl sulphate (IS) and p-cresol sulphate (PCS), which are two protein-bound uraemic toxins originating from the intestinal tract following the metabolism of aromatic amino acids by the intestinal flora, inhibit endothelial proliferation, which may contribute to endothelial cell dysfunction [10,11]. The accumulation of IS might contribute to the progression of CKD by promoting glomerulosclerosis, oxidative stress, and interstitial fibrosis with loss of nephrons [12,13]. IS was involved in the pathogenesis of CVD in CKD including atherosclerosis, peripheral arterial disease, congestive heart failure, vascular access thrombosis, and arrhythmia [14,15,16,17,18]. Clinical research in CKD has also confirmed that these two toxins were valuable surrogate markers of infection, cardiovascular events and all-cause mortality [19,20,21,22]. Moreover, fibroblast growth factor 23 (FGF23), a phosphaturic factor secreted from bone, increased in renal failure [23]. One report showed IS correlated independently with FGF23 in worsening CKD [24]. Evidence has demonstrated that FGF 23 is not just a major regulator in mineral-bone disorder but rather a factor responsible for cardiovascular morbidity and mortality in patients with CKD [25].

Thus, lower serum IS and PCS levels may retard the progression of CKD. AST-120, an orally administered intestinal sorbent that has been available in Japan since 1991, is an agent that could effectively lower serum and urine IS levels and delay the progression of CKD, by reducing the inflammation gene expression in CKD patients [26]. In an initial randomized, double-blind, placebo-controlled trial in America, AST-120 was associated with a significant dose-dependent reduction in serum IS levels and a decrease in uraemia-related malaise [27]. Clinical studies in CKD have demonstrated that decreased IS levels was associated with cardiovascular protective effects [16,28]. Thus, the removal of IS by AST-120 not only ameliorated the progression of renal function, but it also lowered the risk of cardiovascular events in CKD patients [29,30,31,32].

However, due to the high production costs of AST-120, it isnot widely used. A new activated charcoal, CharXgen, extracted and produced from local bamboo in Taiwan, has a porous structure. However, its safety, renoprotective effect, and whether CharXgen could reduce serum biochemistry is unclear. As such, the aim of our study isto investigate the effect CharXgen on clinical parameters in a CKD animal model.

## 2. Results

### 2.1. Material Property

The cross-section and longitudinal section of bamboo charcoal and CharXgen analyzed by scanning electron microscope (SEM) areshowed in Figure 1. Brunauer–Emmett–Teller (BET)(m^2^/g) value of bamboo charcoal and CharXgen was 195.82, 534.39 m^2^/g, respectively. It indicated that the specific surface area was significantly increased after an activation process for bamboo charcoal.

The structure change of bamboo charcoal and CharXgen analyzed by X-ray diffractometer (XRD) is shown in Figure 2A. They have similar peak between 20 to 40° and 40 to 50°. This indicates that the structure was not different before and after activation. Moreover, Figure 2B shows the change of functional group on surface of AST-120, bamboo charcoal and CharXgen measured by Fourier-transform infrared spectroscopy (FTIR). Some functional groups were created as bamboo charcoal was activated. The wave morphology of CharXgen was similar to AST-120.

### 2.2. In Vitro Binding Assay

In this in vitro study, the binding capacity of CharXgen to uraemic substances including potassium dihydrogen phosphate, indole, and p-cresol was evaluated and compared to the capacity of AST-120. The binding affinity to indole was higher in CharXgen compared to AST-120 in conditions such as in small and the large intestine (*n* = 3) (*p* < 0.05, *p* < 0.01, respectively) (Figure 3A). The p-cresol binding affinity to CharXgen was significantly stronger than to AST-120 in condition with small intestine (*p* < 0.05). However, no significant difference was noted in the p-cresol binding affinity between CharXgen and AST-120 in condition with large intestine (Figure 3B). As for the binding affinity to phosphate, an in vitro study mimicking the mouth, stomach, small and large intestine was conducted. AST-120 had a stronger phosphate binding capacity than CharXgen in condition with stomach and small and large intestine (*p* < 0.01, *p* < 0.01 and *p* < 0.05, respectively). Moreover, the binding capacity of CharXgen was similar to AST-120 in condition with mouth, as shown in Figure 3C.

### 2.3. Animal Study

23 rats were divided into five groups—normal (*n* = 4), sham (*n* = 5), CKD (*n* = 5), CKD + AST-120 (*n* = 4) and CKD + CharXgen (*n* = 5). Blood biochemical indicators in each group are shown in Table 1. The data is expressed as the mean ± SD. The systolic blood pressure in each group (normal, sham, CKD, CKD + AST-120, CKD + CharXgen) was 120.4 ± 12.2, 115.6 ± 9.2, 188.1 ± 24.7, 169.2 ± 15.1, and 165.4 ± 7.6 mmHg, respectively. Figure 4 shows the status of the intestinal tract and the internal organ of study of rats in each group 8, 16, and 24 h after feeding with charcoal, AST-120 and CharXgen, respectively. The colorresults of the intestinal tract in CharXgen group were similar to the AST-120 group and not significantly blackening as compared to charcoal.

Liver function, including glutamic oxaloacetic transaminase (GOT) and glutamic pyruvictransaminase (GPT), of the CKD + AST-120 and CKD + CharXgen groups did not appear to have a significant change compared to the normal or sham groups (*p* > 0.05 for all, Figure 5 A,B). The serum albumin results in the five groups aredisplayed in Figure 5C, and the levels were normal in the CKD and CKD + CharXgen groups, 3.9 ± 0.3 and 3.7 ± 0.7 g/dL, respectively, indicating that the albumin level was not significantly different among these groups. Figure 6 displays the renal function observed in the rats from each group. The blood urea nitrogen (BUN) levels in the normal, CKD, CKD + CharXgen and CKD + AST-120 groups was 11.7 ± 1.2, 62.7 ± 13.6, 42.7 ± 5.0, and 54.8 ± 24.2 mg/dL, respectively, and the creatinine (Cr) levels in these groups was 0.4 ± 0.0, 1.3 ± 0.2, 1.0 ± 0.1, 1.1 ± 0.3 mg/dL, respectively. These results gave an indication of renal function—the BUN and Cr were significantly increased in the CKD group (*p* < 0.01, *p* < 0.01) in comparison to the normal or sham groups and significantly decreased in the CKD + CharXgen (*p* < 0.01, *p* < 0.01, respectively) as compared to the CKD group. There were no differences between the CKD and CKD + AST-120 groups. The effect of CharXgen and AST-120 on serum IS and PCS are shown in Figure 7. The serum IS and PCS level in the normal, CKD, CKD + CharXgen, and CKD + AST-120 groups were 1.2 ± 0.3, 8.9 ± 4.5, 3.1 ± 1.5, and 3.0 ± 0.9 and 0.9 ± 0.8, 10.9 ± 4.5, 2.8 ± 1.2, and 4.3 ± 1.2 mg/L, respectively. Serum IS and PCS were higher in the CKD group and lower in rats treated with CharXgen (*p* < 0.05, *p* < 0.01, respectively) and AST-120 (*p* < 0.05, *p* < 0.05, respectively) as compared to the CKD group. Serum calcium and phosphate were not significantly different among the control and study groups (Figure 8 A,B). The FGF23 level in the normal and sham groups was 432 ± 31.1, 427 ± 61.1 pg/mL, respectively, and it was as high as 578 ± 27.8 pg/mL in the CKD group. In the CKD + CharXgen group, FGF23 was significantly decreased to 473 ± 53.0 when compared to the CKD group (578 ± 27.8 pg/mL, *p* < 0.01). There was no difference in the FGF23 level between the CKD + AST-120 (622 ± 98.6 pg/mL) and the CKD group (Figure 8C).

## 3. Discussion

Our results demonstrate that the new activated charcoal, CharXgen with high specific surface area, issafe and non-toxic in vivo. In addition, these findings further support the potential benefit of CharXgen in renal function protection and the reduction of serum FGF23, IS and PCS in uraemic rats. In CKD, metabolic changes and an impaired urinary excretion of metabolites lead to the accumulation of uraemic toxins in the bodies [33]. The accumulated gastrointestinal tract-derived uraemic toxins were regarded as an important risk factor of kidney deterioration and cardiovascular disease in CKD [12,13,20,21,22]. Thus, evidence indicates that IS and PCS are not only vascular toxins but are also nephrotoxins. Because of their high affinity to serum albumin, the removal of IS and PCS is highly dependent on renal function. Our previous research demonstrated that in addition to the kidneys, the liver was an essential and independent organ in determining serum IS and PCS levels [34]. One report by Aronov et al. confirmed the colonic origin of IS and PCS by comparing Hemodialysis patients with and without a colon [35]. Thus, the gut-liver-kidney axis is associated with serum levels of IS and PCS.

AST-120, an oral sorbent from Japan, has been shown to be effective in delaying the progression ofnot only CKD, but also CVD events in patients with advanced CKD by adsorbing the IS precursor, indole, in the intestine, and subsequently reducing serum IS levels [29,30,31,32]. However, the structure of bamboo charcoal, which CharXgen is extracted and produced from, is activated and disintegrates at a high temperature, and eventually becomes more porous and forms more functional groups. These functional groups on the surface of CharXgen can absorb more toxins. Though CharXgen and AST-120 were originated from different production methods, they share some similar structures. For traditional activated charcoal, it was not suitable for long term use in absorbing uremic toxins because of its side effects including black stools, constipation or even blockage of the intestinal tract [36]. The darker the colorof the intestines may reflect the higher the dose of activated charcoal taking. Patients with high dose activated charcoal will be at high risk in leading to serious intestinal complications. Our results showed the bowel color in CharXgen group was similar to the AST-120 group without causing the intestinal to turn black as activated charcoal group. Whether CharXgen has similar characteristics to AST-120 was unknown and this question can be answered in this study. From our in vitro results, CharXgen had a stronger binding capacity for indole and p-cresol than AST-120 in the intestine. However, as for phosphate absorption, AST-120 was stronger than CharXgen. In the animal model, serum levels of IS and PCS were lower in rats fed with CharXgen or AST-120 as compared to CKD rats. These results showed that the binding capacity of CharXgen to protein-bound uraemic toxins was not inferior to AST-120. Concurrently, the improvement of renal function was more significant in the CharXgen group than the AST-120 group.

Moreover, it is worth mentioning that CharXgen significantly lowered FGF23 levels, which is markedly elevated in CKD [37], and elevated FGF23 levels are associated with CVD events, left ventricular hypertrophy [38,39] and with increased progression of CKD [25,40]. Serum FGF23 concentrations have been shown to correlate positively with serum phosphate levels [41]. Recent study showed that the control of serum phosphate could reduce FGF23 levels [42]. From our results, it showed that CharXgen and AST-120 were capable of binding phosphate in conditions such as in small and the large intestine. The average serum phosphate levels of rats in AST-120 group were similar to CKD group, and were lower in CharXgen than in CKD group, though there was no significant difference among these groups. It seemed that the effect of decreased FGF23 in CharXgen group cannot be fully explained by lowering serum phosphate. In addition, experimental study indicated that inflammation was novel regulator of systemic FGF23 synthesis and secretion [43,44]. Previous report demonstrated IS was able to induce oxidative stress and inflammation [11]. In this study, IS levels were significant lower in CharXgen than in CKD group. We speculated that through absorbing IS by CharXgen may be a potential pathway to lower circulating FGF23 levels in CharXgen group. Based on previous studies, the accumulated serum IS was an adverse factor leading to an increased progression of kidney disease [12,13]. Lowering serum IS, PCS, and even FGF23, is an essential way to deter the progression of renal failure. Currently, AST-120 is the most powerful medicine that can decrease serum and urine levels of IS. However, the results from this study showed that CharXgen and AST-120 had similar capacities in reducing the levels of the protein-bound uraemic toxins, IS and PCS. Additionally, CharXgen seemed more likely than AST-120 to maintain renal function.

As for safety, CharXgen neither affected serum albumin nor caused the intestine to darken as does traditional charcoal. Furthermore, liver function did not significantly change after feeding with CharXgen. Thus, these results further demonstrated the safety and potential benefit of CharXgen. There are some limitations of this study. First, we only examined the binding capacity of CharXgen and AST-120 on clinically independent parameters in vitro and in a CKD animal model. Whether CharXgen could delay renal function deterioration or lower serum protein-bound uraemic toxins and FGF23 in humans was not confirmed in this study. Second, we selected indole (precursor of IS) and p-cresol (prototype of PCS) as study candidates for in vitro binding assays, not IS or PCS as was used CKD rats. Third, why only CharXgen not AST-120 reducing serum FGF23 levels, and the exact mechanism remains unclear. Thus, further prospective investigations are required to clarify these intriguing questions.

## 4. Material and Methods

### 4.1. Preparation of CharXgen

CharXgen is a new activated charcoal that was extracted and produced from local bamboo charcoal in Taiwan, which is in a subtropical region and is suitable for bamboo growth. To solve the problem of activated carbon easily adhering to the gastrointestinal wall, we used negatively charged, biocompatible sodium alginate as a coating material, which allowed the activated carbon to form microspheres, preventing it from adhering to the mucosal wall when entering the intestinal tract. The interior of bamboo charcoal is porous, and these holes are very helpful for the adsorption of toxins. The bamboo charcoal can be activated at a high temperature, and its structure will disintegrate, became more porous, and finally, functional groups such as COOH and OHare then generated. The functional groups on the surface of the new activated charcoal can bond with toxic substances.

### 4.2. Physical Property of Materials

The physical character of bamboo charcoal was analyzed before and after activation procedure. The structure change of bamboo charcoal and CharXgen was analyzed by XRD. The change of functional group on surface of AST-120, bamboo charcoal and CharXgen was analyzed by FTIR. BET (m^2^/g) of bamboo charcoal and CharXgen was also calculated under using SEM.

### 4.3. In Vitro Binding Assay

To test the binding affinity of CharXgen to other substances, in vitro experiments simulating gastrointestinal conditions, with different pH levels and incubation times, (mouth: pH 6.5, 5 min, stomach: pH 2, 2 h, small intestine: pH 7.5, 5 h, large intestine: pH 8.0, 24 h) were created. The test substances, including potassium dihydrogen phosphate (KH_2_PO_4_), indole, p-cresol, and a buffer solution were added to a test tube to maintain a stable pH value. Subsequently, CharXgen or AST-120 was added to the test tube. The concentration of the test substance was calculated before and after incubation with AST-120 or CharXgen. The absorption ratio (**Q_e_**) was calculated and obtained from the formula as follows: **Q_e_ =** (**C_0_ − C_e_**) **× V/W,** where C_0_ is the initial concentration, C_e_ is the equilibrium concentration, V is the volume of solution and W is the weight of CharXgen or AST-120.

### 4.4. Color of Bowel in Rats

The intestinal tract and the internal organ of rats in each group at 8, 16, and 24 h after feeding with charcoal, AST-120 and CharXgen, respectively, were removed for further analysis. The qualitative comparison with the change of bowel colorwas completed in each group.

### 4.5. CKD Rat (5/6 Nephrectomy)

Six-week-old male Sprague-Dawley rats (weight 160–180 g) were purchased from Lasco Taiwan Inc. At 8 weeks of age (weight 210–230 g), the rats were anesthetized with Zoletil 50/Xylazine (6 mg/1.3 mg/kg body weight), and the upper and lower thirds of the right kidney were removed. One week after the first operation, the left kidney was removed, leaving approximately 1/6 of the total renal mass. Sham operated rats were used as controls. Two weeks after the 5/6 nephrectomy, the rats were randomized to receive AST-120 (5/6-nephrectomy + AST-120), CharXgen (5/6-nephrectomy + CharXgen) or no treatment (5/6-nephrectomy) for 12 weeks. AST-120 (Kremezin, Kureha Pharmaceuticals, Tokyo, Japan) and CharXgen were administrated post-operatively in the chow at 8% w/w. All animals were divided into five groups: normal, sham, CKD, CKD + AST-120, and CKD + CharXgen. The bowel status was evaluated for rats in all groups 24 h after feeding with AST-120 or CharXgen.

The local committee for the care and uses of laboratory animals at the Mackay Memorial Hospital approved this study (MMH-A-S-108-12). In this study, 5 of 28 rats died after surgery, while the other rats survived until the end of study. All experimental procedures performed on the animals were performed according to Institutional Animal Care and Use Committee (IACUC) policy. All rats were monitored every day and were treated with meperidine (0.3–0.5 mg/100 g body weight for pain control, if needed, based on the physiological parameters as follows: 1. Avoidance, vocalization, and aggressiveness (mainly if the animal cannot escape); 2. Spontaneous activities are reduced, and the animal is isolated from the social group; 3. Altered gait; 4. Hunched posture; 5. Piloerection; 6. Reduced grooming and dark-red stain around the eyes and at nostrils; 7. Reduced appetite and subsequent weight loss; 8. Increased respiration rate; 9. Failure to explore the cage when disturbed. We followed the IACUC policy by using CO_2_ euthanasia as the method of sacrifice at the end of study.

### 4.6. Laboratory Assessment

The following data was collected: albumin (g/dL), BUN (mg/dL), Cr(mg/dL), GPT (IU/L), GOT (IU/L), calcium(mg/dL), phosphate (mg/dL), IS (mg/L), PCS (mg/L), and FGF23 (pg/mL) levels. The bromocresol green method was used to determine albumin levels. FGF23 was measured using a rat ELISA kit (KAINOS Laboratories, Inc., Tokyo, Japan). 

### 4.7. Measurement of IS and PCS

Serum IS and PCS were measured with LC-MS/MS (4000 QTRAP, Framingham, MA, USA). In brief, serum samples were prepared and deproteinized by heat denaturation. The free concentrations of IS and PCS were measured in serum ultrafiltrates, obtained using Microcon YM-30 separators (Millipore, Billerica, MA, USA). HPLC was performed at room temperature using a dC18 column (3.0 × 50 mm, Atlantis, Waters, New Castle, DE, USA). The buffers used were:(A) 0.1% formic acid and (B)1 mM NH_4_OAc + 0.1% formic acid in 100% acetonitrile. The flow rate was 0.6 mL/min with a 3.5-min gradient cycling from 90% A/10% B to 10% A/90% B. Under these conditions, both PCS and IS were eluted at 2.73 and 2.48 min, respectively. Standard curves for PCS and IS were set at 1, 5, 10, 50, 250, 500, and 1000 μg/L. The serum samples for both PCS and IS were processed in the same manner, and they correlated with the serum samples with average r^2^ values of 0.996 ± 0.003. These samples were diluted if the IS or PCS concentrations exceeded the standard curve. Quantitative results were obtained and calculated in terms of their concentrations (mg/L). The sensitivity of this assay was 1 μg/L for PCS and 1 μg/L for IS.

### 4.8. Statistical Analysis

The data is expressed as the mean ± standard deviation (SD). One-way ANOVA, with Tukey’s post hoc test, was used to compare the differences between the control and study groups in the animal model. A *p*-value of less than 0.05 was statistically significant. All statistical analyses were conducted using the SPSS ver. 21.0 software program (IBM, Armonk, New York, NY, USA).

## 5. Conclusions

In conclusion, our study indicated that the novel activated charcoal product, CharXgen, which is extracted from Taiwan local bamboo and has a unique and porous micro-structure, is safe and non-toxic. In vitro and in vivo results further support that CharXgen might have the potential to attenuate renal function deterioration and reduce, not only IS and PCS, but also FGF23 levels in rats with CKD. The protective effect of CharXgen on human kidneys still needs to be confirmed in future investigations.

## Figures and Tables

**Figure 1 ijms-21-01257-f001:**
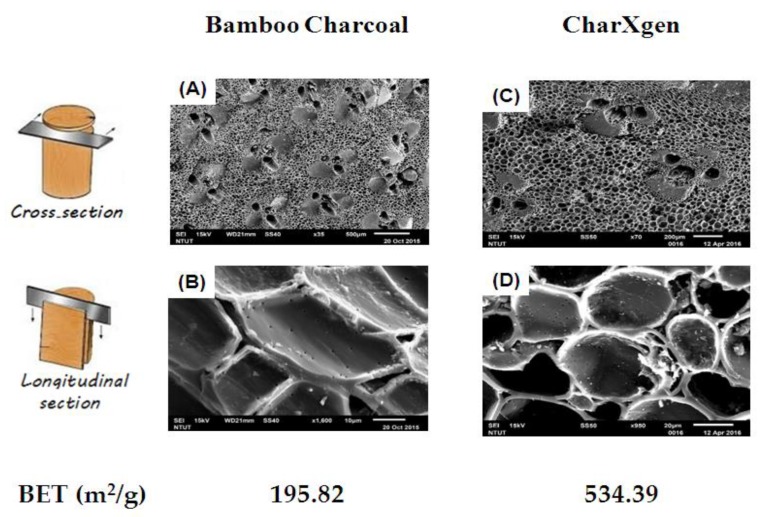
The cross-section and longitudinal section of bamboo charcoal (**A**,**B**) and CharXgen (**C**,**D**) analyzed by scanning electron microscope (SEM). Brunauer–Emmett–Teller (BET) value was significantly increased after an activation process for bamboo charcoal. Scale bar = 100 μm, 10 μm, 200 μm, 20 μm in A, B, C and D, respectively.

**Figure 2 ijms-21-01257-f002:**
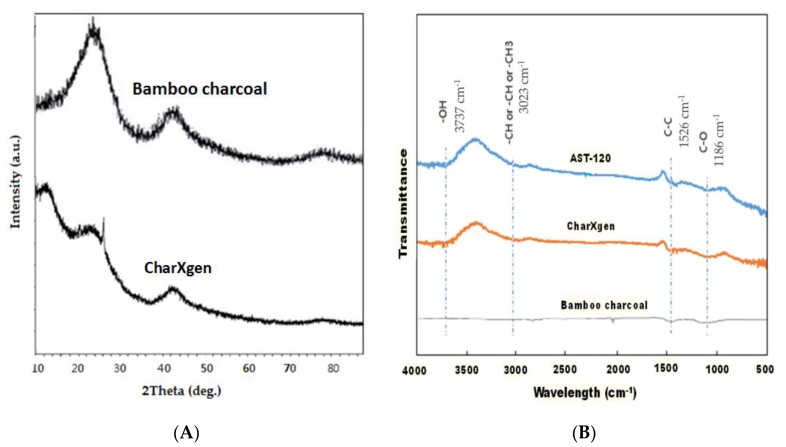
The characteristics of CharXgen, bamboo charcoal, and AST-120. (**A**) The X-ray diffractometer (XRD) of bamboo charcoal and CharXgen. They have similar peak between 20 and 40 degrees and 40 and 50 degrees. It indicated that the structure was not different before and after activation. (**B**) Some functional groups were created as bamboo charcoal was activated by Fourier Transform Infra Red Spectrometer (FTIR). The wave morphology of CharXgen was similar to that ofAST-120.

**Figure 3 ijms-21-01257-f003:**
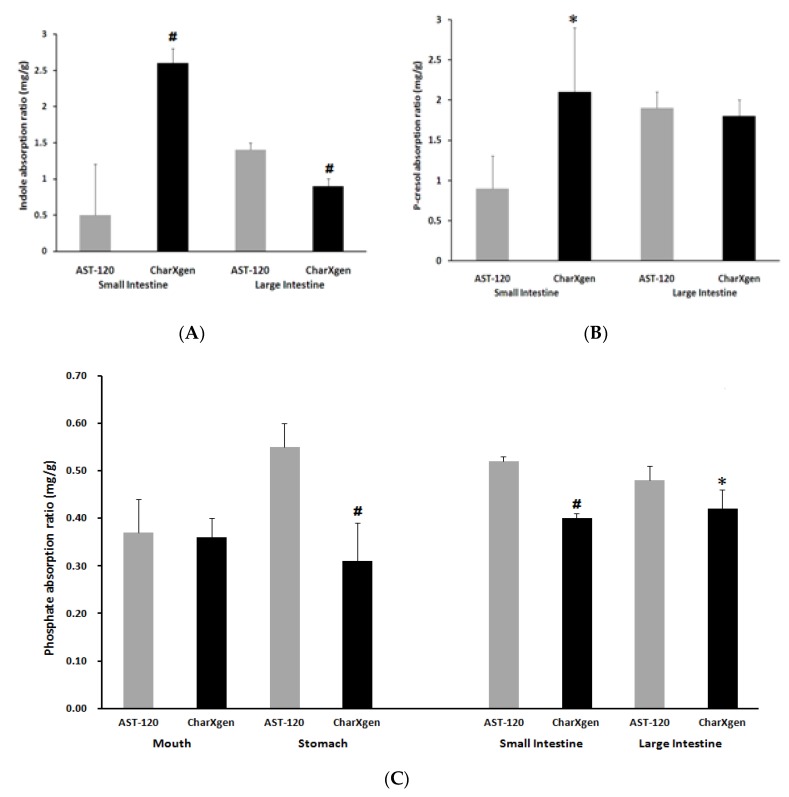
In vitro binding capacity assay for indole (**A**), p-cresol (**B**) and phosphate (**C**) in a gastrointestinal simulation state. ^#^
*p* < 0.01; * *p* < 0.05, compared with the AST-120 group (*n* = 3).

**Figure 4 ijms-21-01257-f004:**
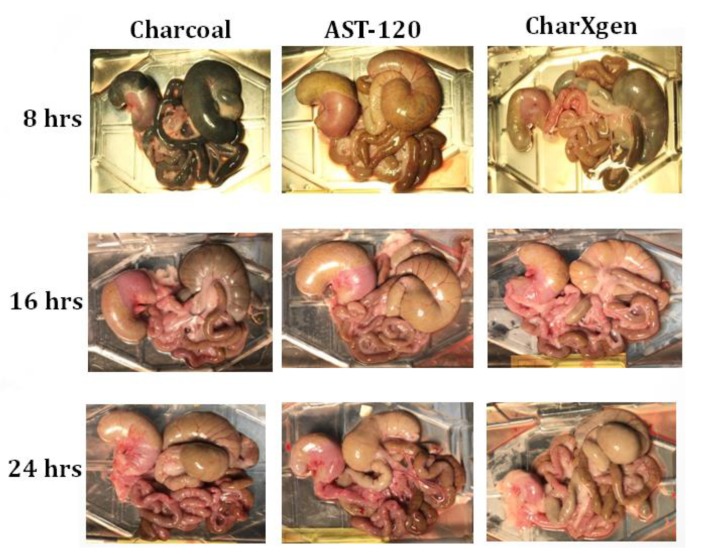
The bowel status in each group of rats 8, 16, and 24 h after feeding with charcoal, AST-120 or CharXgen, respectively.

**Figure 5 ijms-21-01257-f005:**
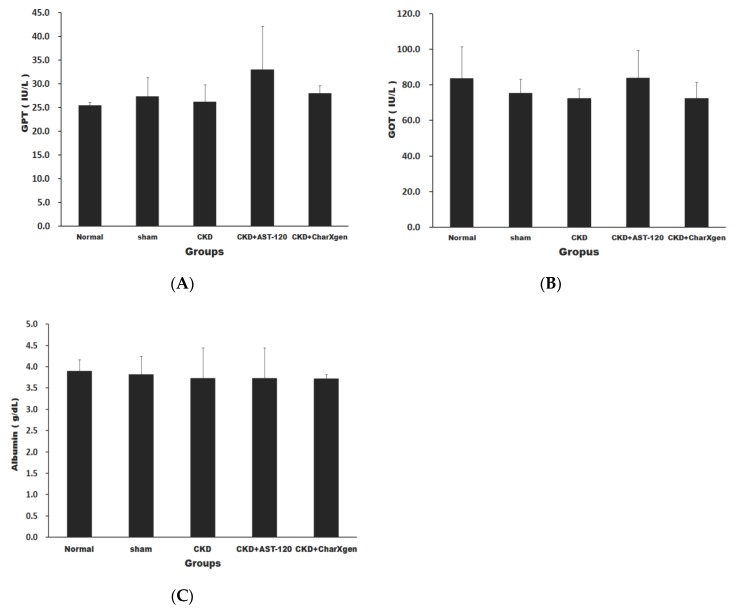
Liver function and nutritional status in the rats from each group. There was no significant difference in GPT (**A**) and GOT (**B**) and albumin (**C**) levels among each group. GPT: glutamic pyruvictransaminase; GOT: glutamic oxaloacetic transaminase.

**Figure 6 ijms-21-01257-f006:**
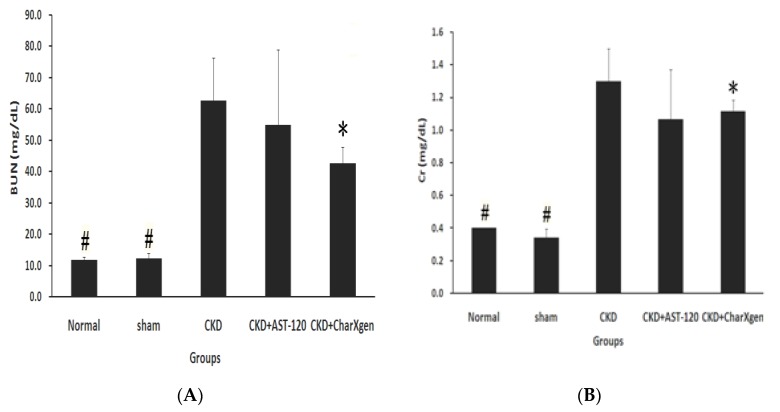
Kidney function in the rats from each group. Serum BUN (**A**) and Cr (**B**) levels were decreased significantly in CKD + CharXgen group as compared to CKD group. ^#^
*p* < 0.01; * *p* < 0.05, compared with the CKD group. BUN: Blood urea nitrogen; Cr: creatinine.

**Figure 7 ijms-21-01257-f007:**
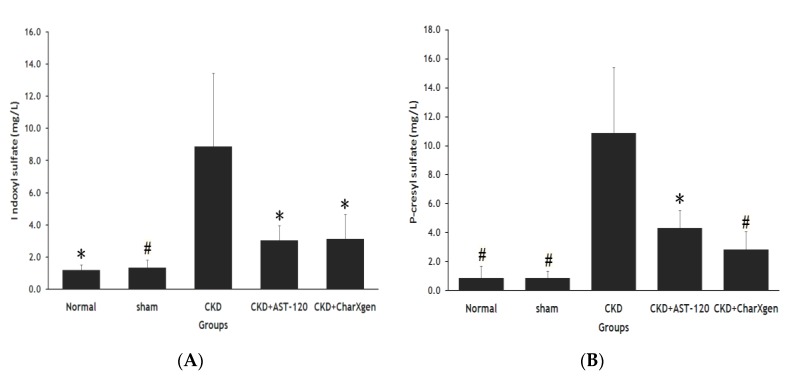
Serum IS (**A**) and PCS (**B**) levels in the rats from each group. Serum IS and PCS levels were significantly lower in CKD + AST-120 and CKD + CharXgen group as compared to CKD group. ^#^
*p* < 0.01; * *p* < 0.05, compared with the CKD group. IS: indoxyl sulphate; PCS: p-cresyl sulphate.

**Figure 8 ijms-21-01257-f008:**
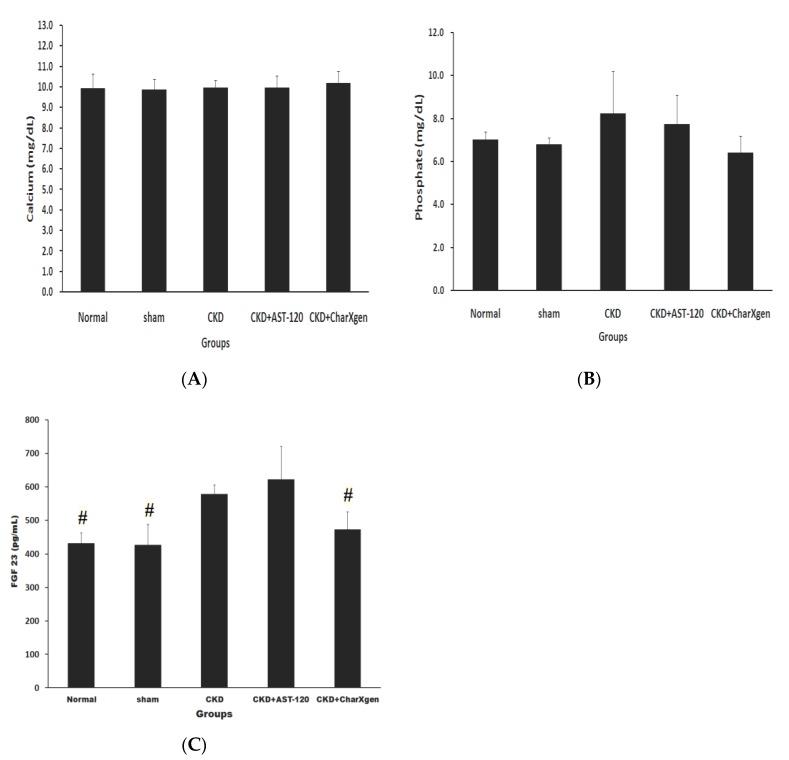
Serum calcium (**A**), phosphate (**B**) and FGF23 (**C**) levels in the rats from each group. There was no significant change for calcium and phosphate among each group. Rats in CKD + CharXgen group had lower FGF23 levels as compared to CKD group. ^#^
*p* < 0.01, compared with the CKD group. FGF23: fibroblast growth factor 23.

**Table 1 ijms-21-01257-t001:** Serum parameters in rats from each group.

Groups	Normal (*n* = 4)	Sham (*n* = 5)	CKD (*n* = 5)	CKD + AST-120 (*n* = 4)	CKD + CharXgen (*n* = 5)
SBP (mmHg)	120.4 ± 12.2	115.6 ± 9.2	188.1 ± 24.7	169.2 ± 15.1	165.4 ± 7.6
Albumin (g/dL)	3.9 ± 0.3	3.8 ± 0.4	3.7 ± 0.7	3.7 ± 0.7	3.7 ± 0.1
GPT (IU/L)	25.5 ± 0.6	27.3 ± 4.0	26.2 ± 3.7	33.0 ± 9.2	28.0 ± 1.6
GOT (IU/L)	83.8 ± 17.6	75.4 ± 7.8	72.4 ± 5.5	84.0 ± 15.5	72.6 ± 9.0
BUN (mg/dL)	11.7 ± 1.2	12.3 ± 1.7	62.7 ± 13.6	54.8 ± 24.2	42.7 ± 5.0
Cr (mg/dL)	0.4 ± 0.0	0.3 ± 0.1	1.3± 0.2	1.1 ± 0.3	1.1 ± 0.1
Ca (mg/dL)	9.9 ± 0.7	9.9 ± 0.5	10.0± 0.3	10.0 ± 0.6	10.2 ± 0.6
P (mg/dL)	7.0 ± 0.4	6.8 ± 0.3	8.2 ± 2.0	7.7 ± 1.4	6.4 ± 0.8
IS (mg/L)	1.2 ± 0.3	1.3 ± 0.5	8.9 ± 4.5	3.0 ± 0.9	3.1 ± 1.5
PCS (mg/L)	0.9 ± 0.8	0.9 ± 0.5	10.9 ± 4.5	4.3 ± 1.2	2.8 ± 1.2
FGF23 (pg/mL)	432.0 ± 31.1	427.0 ± 61.1	578.0 ± 27.8	622.0 ± 98.6	473.0 ± 53.0

The data is expressed as the mean ± SD. SBP: systolic blood pressure; BUN: Blood urea nitrogen; Cr: creatinine; GPT: glutamic pyruvic transaminase; GOT: glutamic oxaloacetic transaminase; Ca: calcium; P: phosphate; IS: indoxyl sulphate; PCS: p-cresyl sulphate; FGF23: fibroblast growth factor 23.

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
