# Peer review of "CharXgen-Activated Bamboo Charcoal Encapsulated in Sodium Alginate Microsphere as the Absorbent of Uremic Toxins to Retard Kidney Function Deterioration"

_ijms, 2020, doi:10.3390/ijms21041257_

Round 1

Reviewer 1 Report

The authors reported the effect of CharXgen on the adsorption of uremic toxins using in vitro and in vivo studies. It is interesting to know the new material for the treatment of CKD compared to AST-120.

In the animal model, the authors concluded that the binding capacity of CharXgen to protein-bound uremic toxins was not inferior to AST-120 while the preservation of kidney function by CharXgen was superior to that with AST-120. Previous studies show that lower indoxyl sulfate induces improvement of kidney function, and they should discuss it more deeply. The reduction of FGF23 is associated with lower phosphorus in CKD patients. The adsorption amount of CharXgen was smaller than that of AST-120 in vitro, and the circulating phosphorus in CKD rat was not different between CharXgen and AST-120 groups. The authors should discuss it more deeply. Line 31 ‘There were no differences in the CKD and the CKD+AST-12o groups’ is not clear. Line 64’ CharXgen had a higher surface area than AST-120’. The authors should put some references. Line 99 ‘AST-120 had a stronger binding capacity than CharXgen in the stomach and small and large intestine. This sentence is not clear. The animal study should show the amount of food intake or nutrition with the administration of CharXgen, and the authors should show the body weight in each group. The authors showed the kidney function, uremic toxins, and mineral parameters both in Table 1 and Figure 5-8. They should discard those data in Table 1.

Author Response

Response to Reviewer comments:

The authors reported the effect of CharXgen on the adsorption of uremic toxins using in vitro and in vivo studies. It is interesting to know the new material for the treatment of CKD compared to AST-120

In the animal model, the authors concluded that the binding capacity of CharXgen to protein-bound uremic toxins was not inferior to AST-120 while the preservation of kidney function by CharXgen was superior to that with AST-120. Previous studies show that lower indoxyl sulfate induces improvement of kidney function, and they should discuss it more deeply.

<Response>
Thanks for this valuable suggestions. We have added more information in the manuscript. Please see the revised manuscript.

The reduction of FGF23 is associated with lower phosphorus in CKD patients. The adsorption amount of CharXgen was smaller than that of AST-120 in vitro, and the circulating phosphorus in CKD rat was not different between CharXgen and AST-120 groups. The authors should discuss it more deeply.

<Response>
Thanks for this valuable comments. We have added some discussions in manuscript.

Line 31 ‘There were no differences in the CKD and the CKD+AST-12o groups’ is not clear.

<Response>
Thanks for this valuable comment. We have revised abstract. The sentence should be changed to “renal function was improved in CKD rats fed with CharXgen as compared to the CKD group, and was no significant differences in the CKD and the CKD + AST-120 groups”.

Line 64’ CharXgen had a higher surface area than AST-120’. The authors should put some references.

<Response>
Thanks for this valuable comments. In our preliminary study, the surface area of AST-120 and Bamboo charcoal was analyzed and compared by SEM. The result showed that Bamboo charcoal had higher specific surface area then AST-120 (data not shown in manuscript) as following figure. CharXgen was subsequently produced from Bamboo charcoal. According to our results (figure 1), CharXgen had higher surface area then Bamboo charcoal. Thus we can speculate that CharXgen had higher surface area then AST-120. However, we agreed that It was not appropriate to put the sentence “CharXgen has a higher surface area than AST-120” in introduction. We have revised the content in introduction.

Line 99 ‘AST-120 had a stronger binding capacity than CharXgen in the stomach and small and large intestine. This sentence is not clear.

<Response>
Thanks for this valuable comment. It should be “AST-120 had a stronger phosphate binding capacity than CharXgen in the stomach and small and large intestine. We have revised the manuscript.

The animal study should show the amount of food intake or nutrition with the administration of CharXgen, and the authors should show the body weight in each group.

<Response>
Thanks for this valuable comments. The daily food intake of rats is fixed, and the research assistant will put a sufficient amount of feed on a stainless steel cage so that rats could eat freely every day. We did not measure the rat body weight in each group. This is also a limitation in the study.

The authors showed the kidney function, uremic toxins, and mineral parameters both in Table 1 and Figure 5-8. They should discard those data in Table 1. 

<Response>
Thanks for this valuable suggestions. We have deleted table 1 for duplicated data presentation. 

Reviewer 2 Report

The authors investigated  the effects of CharXgen – an activated charcoal, on the absorption of uraemic toxins in the bowel and clinical parameters in experimental CKD and compared its properties with another intestinal sorbent – AST-120. This manuscript is interesting, but there are many doubts that should be clarified. Below are my detailed remarks:

Abstract - line 36 should be "lowering", not "lowing" The rationale for studying of FGF23 after CharXgen treatment should be clearly explained in the introduction. The abbreviations like SEM or BET should be explained when used for the first time in the main text Why AST-120 was not analyzed by SEM for comparison to CharXgen? In the introduction authors wrote that CharXgen has a higher surface area than AST-120, but there is no reference or experimental data that prove this statement. When describing the results of in vitro binding capacity it should be clearly stated “in conditions such as in the small intestine” not “in the small intestine“ etc. The significance of changes of parameters in table 1 are not indicated, so it is not known if these changes are significant. Figures 5-8 repeat the data from the table 1. These figures should be removed, or alternatively, the repeated data from table 1 can be removed and presented on figures. There's also no need to repeat the data from table 1 in the text. Does the bowel color after AST-120 or CharXgen has any significance (figure 4)?  If so, then it should be explained. From how many rats the laboratory assessments were made? Were the rats included, from which the bowels were removed after 8, 16 and 24 hrs of charcoal, AST-120 or CharXgen administration,  which must have been related to the euthanasia of the animals? (Figure 4). It is not clear from “methods”. Line 284 – To my knowledge Biocompare is a tool used for comparison of kits and reagents from different manufacturers, not an manufacturer. The references are not very current - only 5 of 31 references are newer than in 2010. Are there any newer papers concerning the subject? In reference 26 correct numbers of pages are 1769-1776

Author Response

Response to Reviewer comments:

The authors investigated  the effects of CharXgen – an activated charcoal, on the absorption of uraemic toxins in the bowel and clinical parameters in experimental CKD and compared its properties with another intestinal sorbent – AST-120. This manuscript is interesting, but there are many doubts that should be clarified. Below are my detailed remarks

Abstract - line 36 should be "lowering", not "lowing"

<Response>
Thanks for this valuable suggestions. We have revised this mistake.

The rationale for studying of FGF23 after CharXgen treatment should be clearly explained in the introduction.

<Response>
Thanks for this valuable suggestion. WE have added more information about FGF23 in introduction.

The abbreviations like SEM or BET should be explained when used for the first time in the main text

<Response>
Thanks for this valuable comment. We have revised the manuscript per your suggestion.

Why AST-120 was not analyzed by SEM for comparison to CharXgen? In the introduction authors wrote that CharXgen has a higher surface area than AST-120, but there is no reference or experimental data that prove this statement.

<Response>
Thanks for this valuable comments. In our preliminary study, the surface area of AST-120 and Bamboo charcoal was analyzed and compared by SEM. The result showed that Bamboo charcoal had higher specific surface area then AST-120 (data not shown in manuscript) as the figure attached. CharXgen was subsequently produced from Bamboo charcoal. According to our results (figure 1), CharXgen had higher surface area then Bamboo charcoal. Thus we can speculate that CharXgen had higher surface area then AST-120. However, we agreed that It was not appropriate to put the sentence “CharXgen has a higher surface area than AST-120” in introduction. We have revised the content in introduction.

When describing the results of in vitro binding capacity it should be clearly stated “in conditions such as in the small intestine” not “in the small intestine“ etc.

<Response>
Thanks for this valuable suggestions. We have revised the manuscript per your suggestion. 

The significance of changes of parameters in table 1 are not indicated, so it is not known if these changes are significant. Figures 5-8 repeat the data from the table 1. These figures should be removed, or alternatively, the repeated data from table 1 can be removed and presented on figures. There's also no need to repeat the data from table 1 in the text.

<Response>
Thanks for this valuable comments. We have deleted table 1 for duplicated data presentation.  

Does the bowel color after AST-120 or CharXgen has any significance (figure 4)?  If so, then it should be explained. From how many rats the laboratory assessments were made? Were the rats included, from which the bowels were removed after 8, 16 and 24 hrs of charcoal, AST-120 or CharXgen administration, which must have been related to the euthanasia of the animals?

<Response>
Thanks for this valuable comments. In figure 4, we only showed the change of bowel color in charcoal, AST-120 and CharXgen group. We did not provide quantitative comparisons between each group. One rat was used in each group and time point (8, 16, and 24 hrs) and total rat numbers were 9.  

(Figure 4). It is not clear from “methods”.

<Response>
Thanks for this valuable comment. We have added some descriptions about figure 4 in method.

 Line 284 – To my knowledge Biocompare is a tool used for comparison of kits and reagents from different manufacturers, not an manufacturer.

<Response>
Thanks for this valuable comment. We have corrected the manufacturers of rat FGF23 ELISA kit

The references are not very current - only 5 of 31 references are newer than in 2010. Are there any newer papers concerning the subject? In reference 26 correct numbers of pages are 1769-1776 

<Response>
Thanks for this valuable suggestions. AST-120 was an old charcoal and has been used more than 20 years. Most about AST-120 and uremic toxins studies were published before. We have renewed some references and corrected the mistake of reference 26.

Round 2

Reviewer 1 Report

The authors have revised their manuscript according to the several comments from the reviewers.

The authors have discussed the results about FGF23 and phosphorus in CKD rats treated with CharXgen or AST-120. However it is still insufficient, and they should discuss it more deeply in the point of mineral bone metabolism, especially FGF23 and phosphorus, because they showed in vitro adsorption study by CharXgen for phosphorus, but not FGF23.

Author Response

Response to Reviewer comments:

Reviewer 1

The authors have discussed the results about FGF23 and phosphorus in CKD rats treated with CharXgen or AST-120. However it is still insufficient, and they should discuss it more deeply in the point of mineral bone metabolism, especially FGF23 and phosphorus, because they showed in vitro adsorption study by CharXgen for phosphorus, but not FGF23. 

<Response>
Thanks for this valuable suggestion. FGF23 levels are markedly elevated in CKD. Serum FGF23 concentrations have been shown to correlate positively with serum phosphate levels. Recent study showed that the control of serum phosphate could reduce FGF23 levels. From our results, it showed that CharXgen and AST-120 were capable of binding phosphate in conditions such as in small and the large intestine. The average serum phosphate levels of rats in AST-120 group was similar to CKD group, and it were lower in CharXgen then CKD group, though there was no significant difference among these groups. It seemed that the effect of decrease FGF23 in CharXgen group cannot be fully explained by lowering serum phosphate. In addition, experimental study indicated that inflammation was novel regulator of systemic FGF23 synthesis and secretion. Previous report demonstrated IS was able to induce oxidative stress and inflammation. In this study, IS levels were significant lower in CharXgen then CKD group. We speculated that through absorbing IS by CharXgen may be a potential pathway to lower circulating FGF23 levels in CharXgen group. The exact mechanism for CharXgen reducing serum FGF23 levels cannot be answered in this study. This is also one of study limitations. We have revised the part of limitations and added more discussion in the manuscript.

Reviewer 2 Report

The paper has been notably improved, but i think that one of my comments was not fully understood. I was asking about the significance of the bowel colour after AST120 or CharXgen treatment. I meant - Is there any clinical significance of the bowel colour? Does darker colour mean some adverse effect on the intestine? This should be discussed.

Author Response

The paper has been notably improved, but i think that one of my comments was not fully understood. I was asking about the significance of the bowel colour after AST120 or CharXgen treatment. I meant - Is there any clinical significance of the bowel colour? Does darker colour mean some adverse effect on the intestine? This should be discussed.

<Response>
Thanks for this valuable comments. Activated charcoal is commonly taken by mouth to treat poisonings at emergency department and is not suitable for long term use in absorbing uremic toxins because of its side effect including black stools, constipation or even slowing or blockage of the intestinal tract <Ref 1,2>. Activated charcoal will be adsorbed by the intestine, causing the intestine to turn black. The darker the color of the intestines may reflect the higher the dose of activated charcoal taking. Patients with high dose activated charcoal will be at high risk in leading to serious intestinal complications. Thus, in order to prevent this side effect, AST-120, a safe activated charcoal, was manufactured through a special process. It was less likely to be adsorbed by the intestine and cause the intestinal color to turn black. In figure 4, we just want to present the change of bowel colour in charcoal, AST-120 and CharXgen group. Our results showed the bowel colour in CharXgen group was similar to the AST-120 group without causing the intestinal to turn black as activated charcoal group. We have added some discussions about this point in manuscript.

<References>

Watson WA, Cremer KF, Chapman JA "Gastrointestinal obstruction associated with multiple-dose activated charcoal." J Emerg Med 4 (1986): 401-7 Atkinson SW, Young Y, Trotter GA "Treatment with activated charcoal complicated by gastrointestinal obstruction requiring surgery." BMJ 305 (1992): 56